# Observational study of effects of HIV acquisition and antiretroviral treatment on biomarkers of systemic immune activation

Ewelina Kosmider[1⊙], Jackson Wallner[2⊙], Ana Gervassi[2], Rachel A. Bender Ignacio[1,3,4], Delia Pinto-Santini[1], German Gornalusse[5], Urvashi Pandey[1,5], Florian Hladik[1,3,5], Paul T. Edlefsen[1], Javier R. Lama[4,6], Ann C. Duerr[1], Lisa M. Frenkel[1,2,3,4,7,8]*

1 Vaccine and Infectious Disease Division, Fred Hutchinson Cancer Center, Seattle, Washington, United States of America, 2 Seattle Children's Research Institute, Seattle, Washington, United States of America, 3 Department of Medicine, University of Washington, Seattle, Washington, United States of America, 4 Department of Global Health, University of Washington, Seattle, Washington, United States of America, 5 Department of Obstetrics & Gynecology, University of Washington, Seattle, Washington, United States of America, 6 Asociación Civil Impacta Salud y Educación, Lima, Perú, 7 Department of Pediatrics, University of Washington, Seattle, Washington, United States of America, 8 Department of Laboratory Medicine and Pathology, University of Washington, Seattle, Washington, United States of America

⊙ These authors contributed equally to this work.
* lfrenkel@uw.edu

**Data Availability Statement:** All relevant data are within the paper and its Supporting information files.

## Abstract

To assess whether biomarkers of systemic inflammation are associated with HIV acquisition or with the timing of ART initiation ("immediate", at diagnosis, versus "deferred", at 24 weeks post-diagnosis) in men-who-have-sex-with-men (MSM) and transgender women, we conducted a retrospective study comparing inflammatory biomarkers in participants' specimens collected before infection and after ≥2 years of effective ART. We measured biomarkers in four longitudinally collected plasma, including two specimens collected from each participant before and two after HIV acquisition and confirmed ART-suppression. Biomarkers were quantified by enzyme-linked immuno-assay or Meso Scale Discovery. When evaluating systematic variation in these markers over time, we found that multiple biomarkers consistently varied across participants' two pre-infection or two post-ART-suppression specimens. Additionally, we compared changes in biomarkers after vs before HIV acquisition. Across 47 participants, the levels of C-reactive protein (CRP), monocyte chemo-attractant protein-1, tumor necrosis factor-α and interferon gamma-induced protein-10 significantly increased while leptin and lipopolysaccharide binding protein (LBP) significantly decreased following HIV infection. Randomization to deferred-ART initiation was associated with greater increases in CRP and no decrease in LBP. Acquisition of HIV appeared to induce systemic inflammation, with elevation of biomarkers previously associated with infections and cardiovascular disease. Initiation of ART during the early weeks of infection tempered the increase in pro-inflammatory biomarkers compared to delaying ART for ~24 weeks after HIV diagnosis. These findings provide insight into potential mediators by which immediate-ART initiation improves health outcomes, perhaps because immediate-ART limits the size of the HIV reservoir or limits immune dysregulation that in turn trigger systemic inflammation.

**Funding:** Funding Statement" to read: "The project was funded by NIH R01 DA040532 (ACD). Additional support came from the Molecular Profiling and Computational Biology Core of the University of Washington and Fred Hutch Center for AIDS Research (award number P30 AI027757), from the Laboratory Core of International Maternal Pediatric Adolescent AIDS Clinical Trials Network (IMPAACT) UM1 AI106716 (Subaward: LMF) and the National Center for Advancing Translational Sciences of the National Institutes of Health under Award Number KL2 TR002317 (GG). The funders had no role in study design, data collection and analysis, decision to publish, or preparation of the manuscript.

**Competing interests:** The authors have declared that no competing interests exist.

## Author summary

A comparison of biomarkers of systemic inflammation were compared across 47 participants' specimens, two collected before infection and two after ≥2 years of effective ART. The levels of C-reactive protein, monocyte chemo-attractant protein-1, tumor necrosis factor-$\alpha$ and interferon gamma-induced protein-10 significantly increased while leptin and lipopolysaccharide binding protein significantly decreased following HIV infection.

## Introduction

The progression of HIV disease to AIDS has been mitigated by antiretroviral treatment (ART) [1–3]. Despite ART, life-threatening non-AIDS-defining comorbidities associated with elevated biomarkers of systemic immune activation occur at increased frequencies in people living with HIV (PWH) compared to uninfected persons [4–14]. While effective ART is associated with normalization of some pro-inflammatory biomarkers, other biomarkers, particularly those associated with monocyte/macrophage activation, remain elevated in PWH compared to uninfected individuals [7]. Studies comparing immune activation biomarkers in PWH versus uninfected persons aim to control for genetic, behavioral, or other pre-existing factors [4, 7, 15]. However, longitudinal studies examining these factors before and following incident HIV infection are lacking. We leveraged specimens collected from individuals with prospectively documented incident HIV infection [16, 17] to compare immune biomarkers in specimens collected prior to HIV infection to those collected after ART suppression of viral replication. Because ART prevents ongoing viral replication but does not prevent infected cells from producing viral proteins and particles, we hypothesized that PWH would demonstrate greater immune activation after ART suppression of viral replication compared to pre-infection values. Additionally, because ART initiated during primary infection limits the size of the HIV reservoir, we hypothesized that PWH initiating ART at diagnosis during acute or early primary infection ("immediate" ART) would show less immune activation compared to those who "deferred" ART for ~24 weeks.

## Methods

Banked specimens from a prospective study of incident HIV infection among seronegative men-who-have-sex-with-men (MSM) and transgender women in Lima, Peru (*Sabes* Study) [16] were utilized to evaluate immune biomarkers. Participants were enrolled into the *Sabes* Study between 7/16/2013 and 7/31/2015 followed through 9/10/2019 (date of last 4th timepoint sample). All participants provided written informed consent including consent for future use of specimens as approved by the Comité Institucional de Bioética de Vía Libre, the Comité Institucional de Bioética de la Asociación Civil Impacta Salud y Educación and the Fred Hutchinson Cancer Center Institutional Review Board. Personal identifiers were retained at the study site in Lima (JRL) for participant tracking and were not available to other co-authors. Participants were screened monthly for HIV acquisition by HIV antibody and nucleic acid amplification testing. The estimated date of detectable infection (EDDI) was calculated as previously described [17, 18]. Following HIV diagnosis, participants were randomized to initiate ART immediately (immediate-ART) or to defer ART for 24 weeks (deferred-ART) [16].

Participants with documented incident HIV infection were selected for this sub-study in February 2020 (with participants data accessed on multiple dates during the preceding months) based on availability of two pre- and two post-infection plasma specimens, with the

rationale to evaluate immune biomarkers in steady-state and avoid transient changes associated with HIV acquisition or ART initiation. Specimens included one plasma sample shortly after enrollment, a second ≤3 months from EDDI (Visits 1 and 2, respectively), a third ≥6 months and a fourth ≥24 months after ART suppression of plasma HIV RNA to <200 copies/mL (ART-suppression) (Visits 3 and 4, respectively).

Biomarkers linked to systemic inflammation and cardiovascular disease were selected for quantification: C-reactive protein (CRP), tumor necrosis factor-α (TNF-α), interleukin (IL-6), soluble urokinase-type plasminogen activator receptor (suPAR), interferon gamma-induced protein 10 (IP-10), interleukin 1β (IL-1β), interleukin 8 (IL-8), interleukin 10 (IL-10) [19–21], lipopolysaccharide binding protein (LBP) [22, 23], and markers associated with bacterial translocation (soluble cluster of differentiation 14 and 163 (sCD14, sCD163)), and antiviral responses (interferon-gamma (IFN-γ), leptin, interleukin α-2a (IFN-α2a), monocyte chemoattractant protein-1 (MCP-1/CCL2)) [24]. Meso Scale Discovery (MSD, Rockville, MD) determined levels of IFN-α2a, IFN-γ, IL-1β, IL-6, IL-8, IL-10, IP-10, leptin, MCP-1/CCL2, TNF-α, LBP, CRP, and ELISAs (R&D Systems, Minneapolis, MN) determined levels of sCD14, sCD163, and suPAR in March/April 2020.

During study analysis, participants' data were accessed on multiple dates in November and December 2020 and throughout 2021 by multiple team members. All biomarker values were $log_{10}$ transformed prior to analysis. For biomarkers at steady state, our sampling design allows for the two pre-infection specimens to be considered as repeated measures for increased precision in estimating a per-person baseline biomarker level. For biomarkers that retain or return to a steady state, the two post-acquisition time points are also considered repeated measures for estimation of that state. To determine the stability of biomarkers across the two pre-infection or two post-suppression time-points, a two-sided, one-sample t-test was used to separately compare values within each pair of pre-infection and post-suppression specimens. We consider a biomarker to be 'unstable' when it significantly differs between the two pre-infection or two post-suppression time-points. Biomarker levels were also compared to physiologic "normal" ranges determined by assay manufacturers or clinical studies (IFN-α2a, leptin) [25, 26] (S1 Table in S1 File). Because a prior study associated elevated plasma sCD14 levels with efavirenz-based-ART [8], we further analyzed values across Visit-3 and -4 in participants switching from efavirenz- to non-efavirenz-based regimens.

To identify biomarkers that differed following HIV infection despite more than six months of ART-suppression, a regression analysis was conducted separately for each biomarker. Participant-specific fixed effects were evaluated across timeframes defined by pre- or post-infection paired visits. Our primary analysis took advantage of the repeated measures sampling design, and descriptive evaluations of the data were consistent with stable variance within markers over time. This effectively allowed us to implement a pooled variance one-sample t-test with repeated measures, treating the paired visits within the pre- or post-infection timeframe as exchangeable. The participant-specific paired values for the t-test are the average of their two pre-infection timepoints and the average of their two post-infection measurements. In addition, we performed a robustness analysis for the unstable biomarkers in which we used Visit 2 to represent a pre-infection measurement and Visit 4 to represent a post-infection measurement. Unadjusted and Holm adjusted p-values and 95% confidence intervals for the change in biomarker value pre- versus post-infection were calculated in R 4.2.1. P-values were Holm adjusted over 15 biomarkers for the analysis on the whole set and over 6 biomarkers for the analysis on the subset of the unstable biomarkers.

Additionally, the regression model included the indicator "deferred-ART" to evaluate whether the duration of infection prior to ART initiation contributed to a difference in biomarkers. P-values of ≤ 0.05 were considered significant. We employed the within-person

variation over the two pre-infection specimens to assess the scale of natural variation and sampling error. We computed the standard deviation (SD) of each biomarker pre-infection and used this for representing changes graphically on a common scale and to aid in interpretation of the magnitudes of changes observed after diagnosis and treatment.

## Results and discussion

### Participants

From among a total of 216 *Sabes* participants with prospectively documented incident HIV infection [16] 50 had specimens from four time-points and fulfilled ART-suppression entry criteria and were used for this study of immune biomarkers. These 50 included 19 participants randomized to immediate-ART and 31 randomized to deferred-ART. Data from all 50 participants were used in assessing the biomarkers' stability and the "per-protocol" analysis. Five participants randomized to deferred-ART initiated ART prior to 24-weeks post-HIV-diagnosis due to low CD4 cell counts or other ART-qualifying events and two of these initiated ART in the immediate-ART timeframe and are included in the immediate-ART group for the "as-treated" analysis; three initiated ART between the immediate- and the deferred-ART timeframes and were excluded from the as-treated analysis. Results of the as-treated analysis are reported in text unless stated otherwise, and the per-protocol analysis is reported in the supplement. The mean interval from EDDI to ART initiation for immediate-ART (N = 21) and deferred-ART (N = 26) groups were 40 (IQR: 24, 50.5) and 209 days (IQR: 199.5, 221.3), respectively (Table 1).

The antiretrovirals provided to all study participants shifted over time: at Visit 3, 43/50 were receiving efavirenz+emtricitabine+tenofovir disoproxil and by Visit 4, 48/50 were receiving elvitegravir/cobicistat+emtricitabine+tenofovir alafenamide. Non-study ART regimens were given to nine participants as medically indicated, including seven protease-inhibitor-based and two efavirenz-based regimens.

### Pro-inflammatory biomarkers

Prior to HIV infection biomarker values for most participants were within established normal ranges (Fig 1 and S1 Table in S1 File). Elevated pre-infection biomarkers, defined as both of a participant's specimens test results above the normal range, included IL-1β (N = 1 participant), IL-6, IFN-γ, and suPAR (N = 2), MCP-1/CCL2, sCD163, and CRP (N = 3), TNFα (N = 9) and/or leptin (N = 23). Most participants' post-infection biomarker levels remained within established norms. Elevated values in both post-infection specimens were observed for IL-1β (N = 1 participant), suPAR and sCD163 (N = 4), IL-6 (N = 5), leptin (N = 6), MCP-1/CCL2 (N = 8), CRP and TNF-α (N = 9) (Fig 1 and S1 Table in S1 File).

Prior to HIV infection the paired pre-infection biomarker levels demonstrated intraparticipant stability, except for significant variability in IP-10, IL-6 and sCD163 (N = 50, S2 Table in S1 File, S1 Fig) but significance was not sustained after Holm adjustment for multiple comparisons. Following ART-suppression, intraparticipant biomarkers demonstrated stability except for sCD163, leptin, IL-8, and LBP (N = 50, S3 Table in S1 File, S2 Fig). sCD163, leptin, and LBP, but not IL-8, sustained significance after Holm adjustment for multiple comparisons.

In the as-treated analysis, comparisons of participant's (N = 47) mean pre-infection biomarker values to their ART-suppressed mean values by a regression analysis detected statistically significant increases in IP-10, MCP-1/CCL2, TNFα, CRP and significant decreases in leptin and LBP (Fig 2), with differences sustained after Holm adjustment for multiple comparisons in all but LBP and TNF-α (S4 Table in S1 File).

**Table 1. Demographic and clinical parameters of participants.**

| Parameter | Per-protocol assignment | As-treated assignment |
|---|---|---|
| | N = 50 | N = 47 |
| **Age at diagnosis; median (IQR years** | 25.6 (21.6, 30.5) | 25.8 (26.1, 30.9) |
| **Sexual orientation/gender identity[a]; N (%)** | | |
| Homosexual/gay male | 29 (58) | 27 (57) |
| Bisexual male | 15 (30) | 14 (30) |
| Transgender female | 6 (12) | 6 (13) |
| **Race/Ethnicity; N (%)** | | |
| Mixed race/Hispanic/Latino | 50 (100) | 46 (100) |
| **Education level; n (%)** | | |
| Primary education or less | 1 (2) | 1 (2) |
| Secondary education | **6 (12)** | 5 (11) |
| Post-secondary education | 43 (86) | 41 (87) |
| **Time interval from EDDI to ART[b] in days; median (IQR)** | | |
| All participants | 184.5 (45, 210) | 189 (42, 214) |
| Immediate group; N = 19 and N = 21 | 42 (24, 51) | 40 (24,50.5) |
| Deferred group; N = 31 and N = 26 | 208 (189, 220) | 209 (199.5, 221.3) |
| | | |
| HIV Viral Load at Diagnosis; Median (IQR) | 5.86 (4.93, 6.86) | 5.83 (4.94, 6.31) |
| (log$_{10}$ copies/ml) | | |
| | | |
| CD4 T cell count at diagnosis; median (IQR) | 444 (305.3, 584) | 444 (340, 576) |

[a] Sexual orientation/gender identity was asked in a single question using common terms in Spanish; all participants were male at birth and reported sex with men, with options translated as heterosexual, bisexual, homosexual, transfeminine, or other/not disclosed

[b] EDDI = estimated date of detectable HIV infection

Comparison of biomarker levels by timing of ART-initiation found a difference in pre-infection to post-ART-suppression by ART timing group for IFN-α2a and CRP), but significance was not sustained after Holm adjustment for multiple comparisons (Fig 2 and S5 Table in S1 File). CRP increased and IFN-α2a decreased in the deferred-ART but not in the immediate-ART group. Both showed significant unadjusted p-value; CRP sustained significance after Holm adjustment (Fig 2, S6, S7 Tables in S1 File). Furthermore, IP-10 and MCP-1/CCL2 increased(both showed significant unadjusted p-value; IP-10 sustained significance after Holm adjustment in both groups, MCP-1/CCL2 sustained significance after Holm adjustment only in the immediate-ART group) and leptin decreased in both groups (sustained significance after Holm adjustment in both groups), while LBP decreased in the immediate-ART group but not in the deferred-ART group; and significance was not sustained after Holm adjustment.

For the six unstable biomarkers we conducted a robustness analysis in which we repeated the regression analysis using only Visit 2 and Visit 4. Visit 2 was defined as a visit within 3 months of HIV EDDI and was chosen to represent a proximal pre-infection state. Visit 4 was chosen to represent a state post-infection and after ≥2 years of ART suppression we expect that the levels of biomarkers will be more stable than at Visit 3.

Comparison of all participant's (N = 47) pre-infection biomarker values at Visit 2 to their ART-suppressed values at Visit 4 detected statistically significant increase in IP-10 and significant decrease in IL-8, with differences sustained after Holm adjustment for multiple

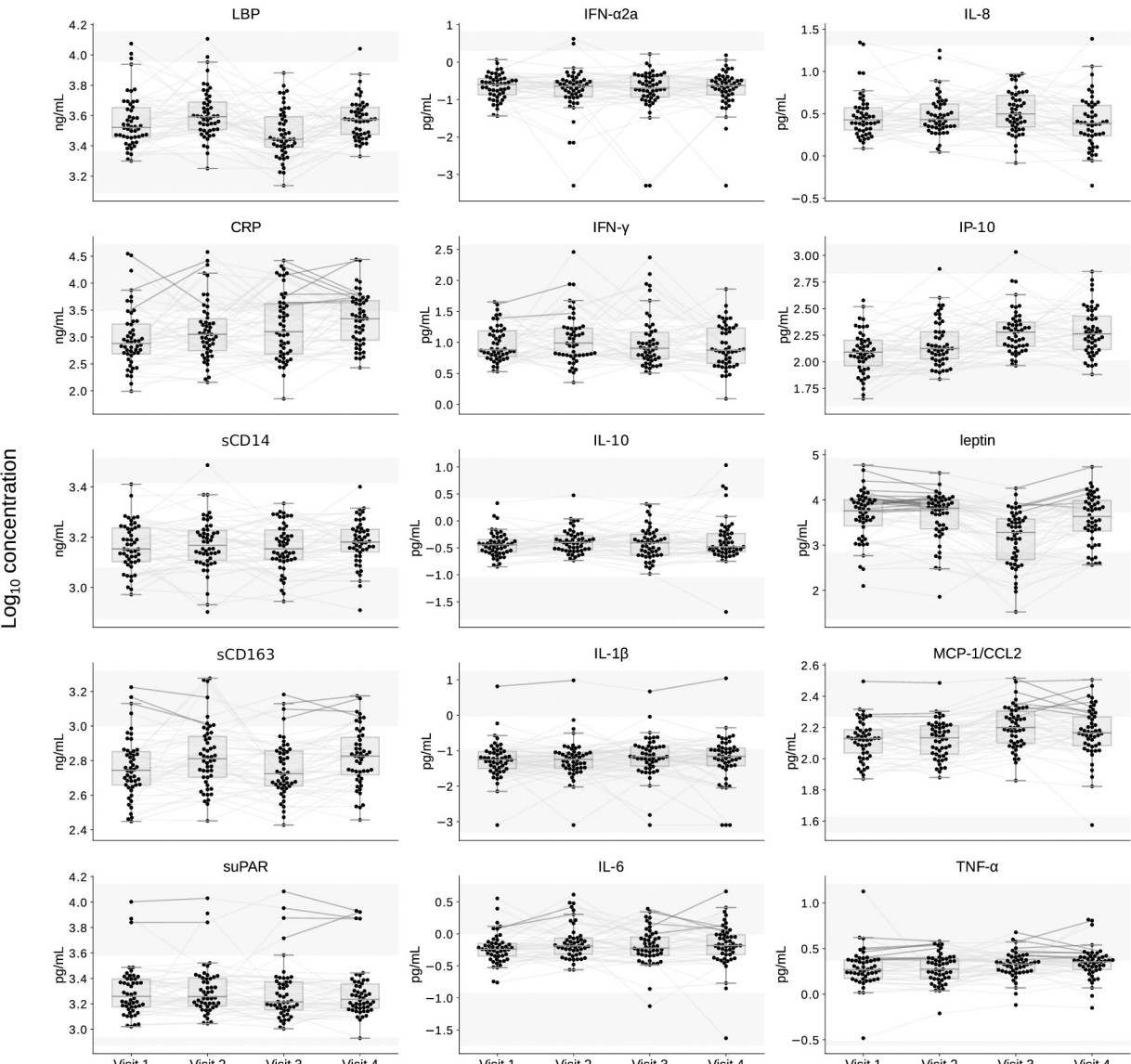

**Fig 1. Plasma biomarker levels.** Each biomarker evaluated is shown in a separate panel, with normal upper and lower ranges indicated between the shaded areas. The biomarker levels are plotted for all participants' (N = 50) two timepoints prior to HIV infection (pre-acquisition), Visit 1 and Visit 2; and two timepoints post-ART-suppression (>6 months and >2 years post ART suppression of plasma HIV RNA to <200 c/mL), Visit 3 and Visit 4. Gray lines connect measurements from the same participant. Lines are bolded if both pre-acquisition or post-ART-suppression values are above the normal range for a given biomarker. Abbreviations: suPAR, soluble urokinase-type plasminogen activator receptor; sCD14 and sCD163, soluble cluster of differentiation 14 and 163; LBP, lipopolysaccharide binding protein; IL-1 β, IL-6, IL-8 and IL-10, interleukin 1b, 6, 8 and 10; IFN-γ and IFN-α2a, interferon-gamma and -alpha 2a; IP-10, interferon gamma-induced protein 10; MCP-1/CCL2, monocyte chemoattractant protein-1; TNF-α, tumor necrosis factor-alpha; CRP, C-reactive protein.

comparisons (S8 Table in S1 File). The increase in IP-10 was consistent with the analysis on the whole set of biomarkers; significant decreases in leptin and LBP were no longer detected; the decrease in IL-8 was only observed using Visits 2 and 4.

Comparison of biomarker levels by timing of ART-initiation did not find any difference in pre-infection to post-ART-suppression by ART timing group for the set of 6 unstable biomarkers (S9 Table in S1 File), which is consistent with the analysis on the whole set.

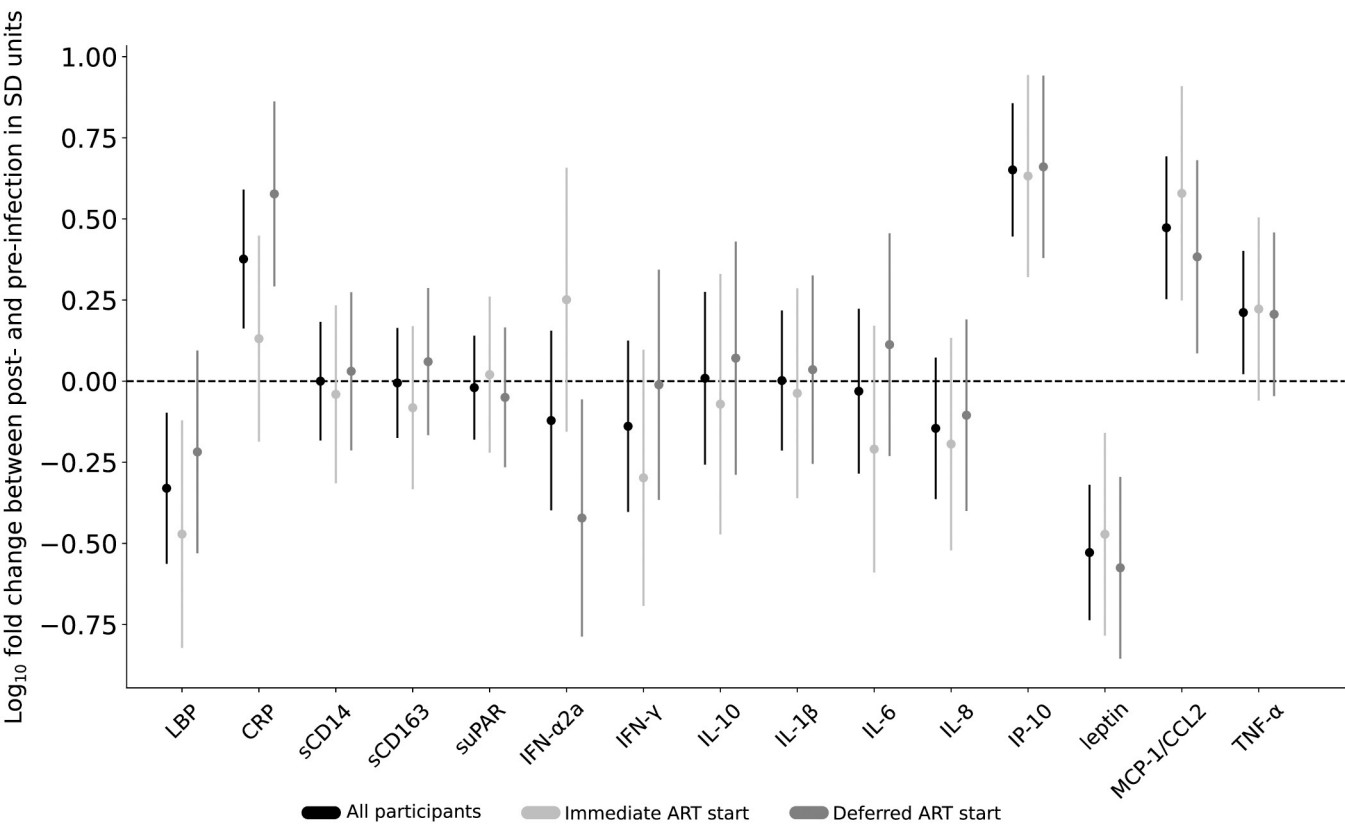

**Fig 2. Difference between pre-HIV-infection and post-ART suppression biomarker values by timing of antiretroviral therapy (ART) initiation.**
Biomarker levels from two specimens before HIV infection and two after ART-suppression were separately compared for all participants (shown in black (N = 47)), and participants separated into two groups: those who started ART immediately upon HIV-diagnosis in light gray (N = 21) vs. those who deferred ART initiation for 24 weeks in dark gray (N = 26), with mean differences (dots) and 95% confidence intervals (CI; lines) shown. The 95% confidence intervals were calculated based on a regression analysis with values $\log_{10}$ transformed prior to analysis and differences divided by pre-infection standard deviation for each analyte. This scaling of each analyte to the z scale facilitates presenting them together on a common figure and has no impact on statistical significance, which is communicated here by the CI not crossing the horizontal dotted line at zero. This line reflects no difference between post- and pre-infection values. Estimates and confidence intervals can be interpreted as a $\log_{10}$ fold-change difference between the two conditions expressed in units of the pre-infection standard deviation of a given analyte (expected fold-change pre-challenge). Abbreviations: suPAR, soluble urokinase-type plasminogen activator receptor; sCD14 and sCD163, soluble cluster of differentiation 14 and 163; LBP, lipopolysaccharide binding protein; IL-1β, IL-6, IL-8 and IL-10, interleukin 1b, 6, 8 and 10; IFN-γ and IFN-α2a, interferon-gamma and -alpha 2a; IP-10, interferon gamma-induced protein 10; MCP-1/CCL2, monocyte chemoattractant protein-1; TNF-α, tumor necrosis factor-alpha; CRP, C-reactive protein.

Levels of IP-10 increased, and levels of IL-8 decreased in the immediate-ART group but not in the deferred-ART group. Both showed significant unadjusted p-value; IP-10 sustained significance after Holm adjustment (S10 and S11 Tables in S1 File). Compared to the analysis on the whole set, significant increase in IP-10 in the deferred-ART group, decrease in leptin in both groups and decrease in LBP in the immediate-ART group were no longer detected; the decrease in IL-8 was only observed using Visits 2 and 4.

Per-protocol analyses of data are shown in S12-S19 Tables in S1 File.

A comparison of sCD14 values between Visit-3 and -4 in participants (N = 43) who switched from efavirenz- to non-efavirenz-based regimens found no significant changes between these intervals.

## Discussion

This study is unique in documenting changes in biomarkers of immune activation in individuals with prospectively documented incident HIV infection, and in examining differences in

biomarkers between participants randomized to initiate ART immediately in early infection or to defer ART for 24 weeks. Comparison of biomarkers from pre-infection to post ART-suppression, including correction for multiple comparisons, showed increased plasma levels of proinflammatory chemokines/cytokines MCP-1/CCL2 and IP-10 secreted by monocyte/macrophages and other cell types in response to HIV infection [27, 28] or stimulation by cytokines [29], and CRP, a marker of inflammation associated with infections, cancers, auto-immunity or tissue damage. These findings of elevated inflammatory cytokines/chemokines are consistent with those reported by previous studies [4, 6, 7, 30].

In addition, the levels of one biomarker, leptin, were lower in the post-infection compared to pre-infection specimens at Visit 3, although by Visit 4 leptin returned to pre-infection levels (Fig 1). The decrease in leptin, a marker of energy expenditure [31], is consistent with previous studies finding lower leptin in ART-treated PWH [32], likely due to HIV-infection-induced catabolism.

We expected that any changes in these biomarkers would be most dramatic when contrasting between pre- and post-infection time ranges, and we expected that within these time ranges the biomarkers would be relatively stable, and thus our primary design was to consider the multiple time points as repeated measures within each time range. However, we found that multiple biomarkers, including IP-10, IL-6, sCD163, leptin, IL-8, and LBP, were significantly changed in the same direction across participants, either within each participant's two pre-infection or two post-ART-suppression specimens. Temporal perturbations in these analytes within individuals may have occurred due to intercurrent illnesses, alcohol consumption [33, 34], or other unknown reasons. However, the finding of common directions of changes across participants suggests a systematic source of variation. While it is not unexpected that HIV infection followed by ART initiation may transiently perturb some of these markers consistently across participants, the finding that there are systematic trends prior to infection in the movement of these biomarkers was not expected when we began this analysis. However, it is consistent with the recently published analysis that we conducted in the larger *SABES* prospective study cohort [35], of which these 50 participants are a subset. In the larger cohort, we observed a systematic trend in some biomarkers that differed between persons who became infected and those who did not become infected [35]. In this analysis, we found biomarkers that differentiated participants at baseline who would ultimately become persons living with HIV from those that ended the trial uninfected, a difference that systematically reduced until there was no significant difference at the pre-infection time point [cite that paper, and maybe say more details such as which markers were consistent across analyses, and if directionality was consistent].

Our comparison of biomarker levels between those who initiated immediate- versus deferred-ART initiation found that those treated immediately had significantly less elevation of their CRP and LBP values. Earlier initiation of ART limits the size of the persistent viral reservoir [36], which should limit production of viral nucleic acids and proteins that others have found associated with progression of carotid artery intima thickness [13] or atherosclerotic plaque [14]. The greater decrease in plasma IFN-α2a observed among those deferring ART (Fig 2) may be attributable to consistently high pre-infection values in this group. Notably, during ART all participants had IFN-α2a values in the normal range of the assay (Fig 1).

While efavirenz in prior studies was associated with elevated sCD14 and kynurenine-tryptophan ratio [8, 37], we did not observe a significant change in sCD14 in participants who switched from efavirenz-based ART to a "non-efavirenz" elvitegravir-based regimen. It is not known whether elvitegravir (or cobicistat, contained in the co-formulated product to reduce hepatic clearance of elvitegravir) is associated with inflammation, but it is notable that sCD14 levels were in the normal range for most specimens tested in the study.

The primary limitations of this study are the relatively small size of the cohort examined, a relatively short follow-up of the ART-suppressed participants for this life-long infection and the assessment of biomarkers from relatively few timepoints. In addition, the relative youth of our study participants and the short duration of their HIV infections limited our ability to observe non-AIDS adverse events, and we were unable to conduct long term follow-up to observe and correlate our findings with clinical events. Additionally, when evaluating the potential impact of efavirenz, we did not test some biomarkers found to be abnormal in other studies, e.g., kynurenine/tryptophan ratio [8, 37]. The primary strength of this study comes from the comparison of two samples from before and two after documented incident HIV infection. The two specimens prior and two after infection reduce variability due to extraneous events. The study of individuals with incident infection diminishes the biases due to pre-existing conditions and confounding behavioral practices, although, we acknowledge that behaviors may change following HIV diagnosis [38].

## Conclusions

In conclusion, multiple pro-inflammatory biomarkers appear to have been induced by HIV despite virologic suppression. Importantly, ART initiation during acute/early HIV infection tempered the increase in pro-inflammatory biomarkers compared to delaying ART for ~24 weeks after HIV diagnosis. These findings provide insight into potential mediators by which immediate-ART initiation improves health outcomes, perhaps because immediate-ART limits the size of the HIV reservoir or limits immune dysregulation that in turn trigger systemic inflammation. Given that systemic inflammation, and in particularly CRP, is a biomarker strongly association with cardiovascular disease [39, 40], these findings provide insight in the mechanisms by which initiation of ART during primary infection in people living with HIV may improve health outcomes.

## Supporting information

**S1 Fig. Difference between two pre-infection time-points.** $\log_{10}$ biomarker levels from two specimens before HIV infection were compared for all participants (n = 50), with mean differences (dots) and 95% confidence intervals (lines) shown. The 95% confidence intervals were calculated based on a two-sided, one-sample t-test with values $\log_{10}$ transformed prior to analysis and differences divided by pre-infection standard deviation for each analyte. The vertical dotted line reflects no difference between two pre-HIV-infection time-points.
(PDF)

**S2 Fig. Difference between two post-infection time-points.** $\log_{10}$ biomarker levels from two specimens after HIV infection were compared for all participants (n = 50), with mean differences (dots) and 95% confidence intervals (lines) shown. The 95% confidence intervals were calculated based on a two-sided, one-sample t-test with values $\log_{10}$ transformed prior to analysis and differences divided by pre-infection standard deviation for each analyte. The vertical dotted line reflects no difference between two post-HIV-infection time-points.
(PDF)

**S1 File.**
(ZIP)

**S1 Checklist. STROBE statement—Checklist of items that should be included in reports of observational studies.**
(PDF)

## Acknowledgments

The authors acknowledge the contributions of the study participants, site investigators and staff; the contribution of the study drugs provided at no cost to the Institution by Gilead Sciences Inc and by Merck Sharp & Dohme Corp. The authors acknowledge that the opinions expressed in this paper are those of the authors and do not necessarily represent those of Gilead Sciences Inc or Merck Sharp & Dohme Corp or the official views of the National Institutes of Health.

## Author Contributions

**Data curation:** Ewelina Kosmider, Jackson Wallner, Ana Gervassi, Rachel A. Bender Ignacio, Delia Pinto-Santini, German Gornalusse, Urvashi Pandey, Florian Hladik, Paul T. Edlefsen, Javier R. Lama, Ann C. Duerr, Lisa M. Frenkel.

**Formal analysis:** Ewelina Kosmider, Paul T. Edlefsen.

**Funding acquisition:** German Gornalusse, Ann C. Duerr, Lisa M. Frenkel.

**Investigation:** Ewelina Kosmider, Jackson Wallner, Ana Gervassi, Rachel A. Bender Ignacio, Delia Pinto-Santini, German Gornalusse, Urvashi Pandey, Florian Hladik, Javier R. Lama, Ann C. Duerr, Lisa M. Frenkel.

**Methodology:** Ewelina Kosmider, Jackson Wallner, Ana Gervassi, Rachel A. Bender Ignacio, Delia Pinto-Santini, German Gornalusse, Urvashi Pandey, Florian Hladik, Paul T. Edlefsen, Javier R. Lama, Ann C. Duerr, Lisa M. Frenkel.

**Project administration:** Jackson Wallner, Ana Gervassi, Delia Pinto-Santini, Ann C. Duerr, Lisa M. Frenkel.

**Writing – original draft:** Ewelina Kosmider, Jackson Wallner, Lisa M. Frenkel.

**Writing – review & editing:** Ewelina Kosmider, Jackson Wallner, Ana Gervassi, Rachel A. Bender Ignacio, Delia Pinto-Santini, German Gornalusse, Urvashi Pandey, Florian Hladik, Paul T. Edlefsen, Javier R. Lama, Ann C. Duerr, Lisa M. Frenkel.

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
