## [Decision Letter · Decision Letter 0]

22 Aug 2023

PONE-D-23-18653Observational study of effects of HIV Acquisition and Antiretroviral Treatment 1 on Biomarkers of Systemic Immune ActivationPLOS ONE

Dear Dr. Frenkel,

Thank you for submitting your manuscript to PLOS ONE. After careful consideration, we feel that it has merit but does not fully meet PLOS ONE’s publication criteria as it currently stands. Therefore, we invite you to submit a revised version of the manuscript that addresses the points raised during the review process.

ACADEMIC EDITOR:

1) Please pay attention to the comments made by reviewer number 2 regarding figure 1, and your statistical analysis procedures.

2) You may include the two supplementary figures within the main text.3) Please provide references for the normal values stated in figure 1. You only provided this information for leptin, CRP and IFN-α2, and not all the other analytes shown in that figure. You may leave these reference/normal values out if no concrete information about them is available.

We look forward to receiving your revised manuscript.

Kind regards,

Novel N. Chegou, Ph.D

Academic Editor

PLOS ONE

“The project was funded by NIH R01 DA040532 (ACD). Additional support came from the Molecular Profiling and Computational Biology Core of the University of Washington and Fred Hutch Center for AIDS Research (award number P30 AI027757), from the Laboratory Core of International Maternal Pediatric Adolescent AIDS Clinical Trials Network (IMPAACT) UM1 AI106716 (Subaward: LMF). This work was also supported by the National Center For Advancing Translational Sciences of the National Institutes of Health under Award Number KL2 TR002317 (GG). The content is solely the responsibility of the authors and does not necessarily represent the official views of the National Institutes of Health.”

“The authors acknowledge the contributions of the study participants, site investigators and staff. The study drug was provided at no cost to the Institution by Merck Sharp & Dohme Corp. The opinions expressed in this paper are those of the authors and do not necessarily represent those of Merck Sharp & Dohme Corp.

The project was funded by NIH R01 DA040532 (ACD). Additional support came from the Molecular Profiling and Computational Biology Core of the University of Washington and Fred Hutch Center for AIDS Research (award number P30 AI027757), from the Laboratory Core of International Maternal Pediatric Adolescent AIDS Clinical Trials Network (IMPAACT) UM1 AI106716 (Subaward: LMF). This work was also supported by the National Center For Advancing Translational Sciences of the National work was also supported by the National Center For Advancing Translational Sciences of the National of the authors and does not necessarily represent the official views of the National Institutes of Health.”

“The project was funded by NIH R01 DA040532 (ACD). Additional support came from the Molecular Profiling and Computational Biology Core of the University of Washington and Fred Hutch Center for AIDS Research (award number P30 AI027757), from the Laboratory Core of International Maternal Pediatric Adolescent AIDS Clinical Trials Network (IMPAACT) UM1 AI106716 (Subaward: LMF). This work was also supported by the National Center For Advancing Translational Sciences of the National Institutes of Health under Award Number KL2 TR002317 (GG). The content is solely the responsibility of the authors and does not necessarily represent the official views of the National Institutes of Health.”

6. Please amend either the title on the online submission form (via Edit Submission) or the title in the manuscript so that they are identical.

Reviewers' comments:

Reviewer's Responses to Questions

**Comments to the Author**

1. Is the manuscript technically sound, and do the data support the conclusions?

Reviewer #1: Yes

Reviewer #2: No

Reviewer #3: Yes

2. Has the statistical analysis been performed appropriately and rigorously? 

Reviewer #1: Yes

Reviewer #2: I Don't Know

Reviewer #3: Yes

3. Have the authors made all data underlying the findings in their manuscript fully available?

Reviewer #1: Yes

Reviewer #2: No

Reviewer #3: No

4. Is the manuscript presented in an intelligible fashion and written in standard English?

Reviewer #1: Yes

Reviewer #2: Yes

Reviewer #3: Yes

5. Review Comments to the Author

Reviewer #1: The present study compared various inflammatory and immune activation biomarkers before HIV acquisition and when participants had a confirmed suppression of viral replication. Several previous studies have evaluated the effects of HIV infection and ART on biomarkers, but the present study provides interesting, previously unpublished information related to those biomarkers. Each participant serves as their own healthy control and that is very interesting and has a great scientific value.

The article is well written, the statistical analysis is well documented and finally results and discussion are clearly presented.

Some minor comments to the authors:

- Pag 8 line 48: could the authors explain why they describe significant intra-participant variability in IP-10, IL-6 and sCD163, when the Holm adjuted p-value is not significant (supplementary table 1)?

- Figure 2, I suggest modifying “differences” to “change” in ordinate axis. I encourage the authors to include significant p-values in the figure.

- In all the supplementary tables, I suggest modifying the order of the columns: first the estimated value of the change, second the p-value and third the Holm adjusted p-value.

- Supplementary Table 4a: How does the author interpret the difference of significance of p-values between p-value and Holm adjusted p-value?

- Page 9 line 191. I think it should be the horizontal dotted line.

Reviewer #2: "Article: Observational Study of the Effects of HIV Acquisition and Antiretroviral Treatment on Biomarkers of Systemic Immune Activation"

In this article, the authors hypothesized that (1) individuals living with HIV (PWH) would demonstrate greater immune activation after ART suppression of viral replication compared to pre-infection values, and (2) PWH initiating ART at the time of diagnosis during acute or early primary infection (referred to as "immediate" ART) would show less immune activation compared to those who deferred ART for approximately 24 weeks.

The authors concluded that certain plasma biomarkers, including CPR, IP-10, MCP-1, and TNFa, were activated, while leptin was inactivated after HIV infection with viral suspension. These results were consistent with previous reports.

As mentioned by the authors, this study is unique as it documents changes in biomarkers of immune activation in individuals with prospectively documented incident HIV infection and examines differences in biomarkers between participants randomized to initiate ART immediately in early infection or to defer ART for 24 weeks. However, some parts of the article, especially the graphs and calculations, are confusing.

Here are the comments:

Major comments

On page 6, lines 129 to page 7, line 135, the description of "Time interval from EDDI to ART in days" in the text and Table 1 is confusing. The reviewer understands that Table 1 describes the "as-treated analysis". Please include the actual numbers for "immediate-ART" and "defer-ART" in the table. Additionally, clarify in the "Method" where the 3 participants who initiated ART between "immediate" and "defer" were analyzed.

The reviewer recommends revising Figure 1 to be a "violin plot" or "boxplot" to better display the distribution of the biomarkers, as it is currently unclear. The authors concluded that CRP, LBP, IP-10, leptin, and TNF-a were significantly different between the "pre-" and "post-" HIV infection stages. However, the plot appears to include each other. Also, in the description of Figure 1, the authors mentioned that "The mean biomarker levels of most participants remained within established norms (page 8, line 154). However, some biomarkers such as suPAR, CRP, IL-6, leptin, and TNF-a appear higher than the normal level. Do the authors have any insight into whether the MSM population shows higher values for these biomarkers? Additionally, does CRP need a symbol for "Holm adjusted p-value <0.001"?

The Y-axis title "Difference between Post- and Pre-infection in SD units" of Figure 2 is confusing. Please explain why the "difference" needs to be "normalized by SD". The reviewer believes that the unit of Figure 2 should be portrayed as a "fold difference". For example, does CRP elevate by ~35% after infection? Please confirm.

In the "Conclusion", the authors suddenly mentioned "Given the strong association of CRP with cardiovascular disease these findings emphasize that HIV prevention and ART initiation during primary infection could diminish non-AIDS events", however, there is no further comments for this, such as this CRP fold change might be read a risk factor for cardiovascular disease. Please comment for this. In addition, what the authors are thinking an application for elevation of IP-10, MCP-1, and TNFa, and also depression of leptin? Or do authors thing these change are clinically relevant?

Minor comments

Page 5, line 87: Did the authors confirm non-HIV infection for Visit 2 samples?

Page 5, lines 97-100: We understand that there is a large inter-analytical variation for biomarker ELIZA analysis. What is the variation of each analyte (was it an average of multiple analyses) for each biomarker? And did this variation affect the final data analyses?

Page 6, lines 120-122: Please explain the utility of "mean/SD" for data analysis. The reviewer believes that "mean/SD" is an indicator of variation, and when it is small, the data variation is also small. However, the authors used it to normalize the magnitude. Please explain the validity of this approach.

Supplemental Table 3: The reviewer believes that the difference between "as-treated" and "per-treated" involves approximately 5 participants, yet some of the biomarkers' estimated fold changes were large. Did the data from these participants highly influence the results? Please provide an explanation. With regard to the "estimated change," does it refer to the fold change of "post-dose" compared with "pre-dose", even if the number was normalized by SD? For example, the estimate of CRP is 0.21, indicating a 21% higher value for the "post-treatment."

Reviewer #3: In the article ‘Observational study of effects of HIV Acquisition and Antiretroviral Treatment 1 on Biomarkers of Systemic Immune Activation’ submitted for review at PLOS One, the authors use a retrospective study comparing inflammatory biomarker measurements collected before and after ≥2 years of effective ART in men-who-have-sex-with-men (MSM) and transgender women in Lima, Peru. The article is well written, easy to follow, and the conclusions seem generally plausible given the results of the statistical analyses. However, I do have a few comments:

Major Comments

1. Page 6, lines 115-116: ‘This effectively allowed us to implement a pooled variance one-sample t-test with repeated measures, treating the paired visits within the pre- or post-infection timeframe as exchangeable.’

- I do have some concerns about the analysis design. Firstly, I do think this part can be strengthened with a better explanation. It is not clear how the paired t-test is conducted between the pre- and post- ART timepoints from this explanation, but looking at Figure 1, it seems that the participant-specific paired values for the t-test are the average of their two pre- ART timepoints and the average of their two post- ART measurement. Can the authors please elaborate more on this in the draft?

Also, with four distinct biomarker measurements, I am curious as to why the authors decide to conduct a pre-post design by pooling the pre- and post- infection timepoints together. Also, the assumption of exchangeability seems to have failed for some of these biomarkers (Supplementary Figures 1 and 2), given that the difference of the two pre- and post- ART measurements are statistically different than 0 for them. In that light, I am wondering why multiple measurement models, like the mixed-effects models or GEE were not pursued here. Also, it may make more sense to pool the two pre-ART timepoints, but given the two post- ART timepoints are 18 months apart, and given that timing of ART and the course of ART over time may have a temporal impact on expression of these biomarkers, what is the plausibility of pooling the two post- ART timepoints together?

Minor Comments

1. Page 5, lines 86-88: ‘Specimens included one plasma sample shortly after enrollment, a second ≤3 months from EDDI (Visits 1 and 2, respectively)’: The second sample was ≤3 months before or after the EDDI? In case it is after, can it be confirmed that it is still a pre-ART timepoint?

2. How is the ‘per-protocol’ analysis defined (as reported in Supplementary Tables 3b, 4b, 5b, and 6b). I don’t think it was mentioned anywhere in the draft.

3. One requirement for the data availability statement is that ‘data available on request from the author’ is not a sufficient response, and if data are indeed only available upon request, the authors must answer ‘No’ for the question - ‘Do the authors confirm that all data underlying the findings described in their manuscript are fully available without restriction?’ In the manuscript information page (Page 4), the authors answered ‘Yes’ to the aforementioned question, but state in data availability statement that ‘Summary data available in the manuscript. Participant level data is stored at Fred Hutch and de-identified data is available by request of Ann Duerr’. This is a contradiction that they need to resolve.

4. Given the number of biomarkers assessed, multiple testing adjustment is indeed necessary to parse out the results, and in this regard, I do appreciate that the authors have put careful thought into this and presented Holm adjusted p-values with their results. In the description of results in Page 9 (lines 173-182), it is important to point out both sets of results. For example, the authors start by stating,

‘Comparisons of all participant’s (N=47) mean pre-infection biomarker values to their ART-suppressed mean values by a regression analysis detected statistically significant increases in IP-10, MCP-1/CCL2, TNFα, CRP and significant decreases in leptin and LBP (Figure 1), with differences sustained after Holm adjustment for multiple comparisons in all but LBP and TNF-α (Supplementary Table 3a).’

This is precisely how these results should be presented in my opinion (with conclusion from the unadjusted and adjusted analyses). However, in the next set of sentences (lines 172-183), only results from the unadjusted analyses are given. Can the authors please add the results from the adjusted analyses as well, as has been done in the first sentence of that paragraph.

6. PLOS authors have the option to publish the peer review history of their article (what does this mean?). If published, this will include your full peer review and any attached files.

Reviewer #1: No

Reviewer #2: No

Reviewer #3: No

---

## [Decision Letter · Decision Letter 1]

15 May 2024

Observational study of effects of HIV Acquisition and Antiretroviral Treatment on Biomarkers of Systemic Immune Activation

PONE-D-23-18653R1

Dear Dr. Frenkel

We’re pleased to inform you that your manuscript has been judged scientifically suitable for publication and will be formally accepted for publication once it meets all outstanding technical requirements.

Kind regards,

Novel N. Chegou, Ph.D

Academic Editor

PLOS ONE

Additional Editor Comments (optional):

Reviewers' comments:

Reviewer's Responses to Questions

**Comments to the Author**

1. If the authors have adequately addressed your comments raised in a previous round of review and you feel that this manuscript is now acceptable for publication, you may indicate that here to bypass the “Comments to the Author” section, enter your conflict of interest statement in the “Confidential to Editor” section, and submit your "Accept" recommendation.

Reviewer #1: All comments have been addressed

Reviewer #3: All comments have been addressed

2. Is the manuscript technically sound, and do the data support the conclusions?

Reviewer #1: Yes

Reviewer #3: Yes

3. Has the statistical analysis been performed appropriately and rigorously? 

Reviewer #1: Yes

Reviewer #3: Yes

4. Have the authors made all data underlying the findings in their manuscript fully available?

Reviewer #1: Yes

Reviewer #3: Yes

5. Is the manuscript presented in an intelligible fashion and written in standard English?

Reviewer #1: Yes

Reviewer #3: Yes

6. Review Comments to the Author

Reviewer #1: The authors have reflected and included in the new version of the paper, all the reviewers comments, improving the quality of the work.

Reviewer #3: (No Response)

7. PLOS authors have the option to publish the peer review history of their article (what does this mean?). If published, this will include your full peer review and any attached files.

Reviewer #1: No

Reviewer #3: No

---

## [Editor Report · Acceptance letter]

20 May 2024

PONE-D-23-18653R1 

PLOS ONE

Dear Dr. Frenkel, 

I'm pleased to inform you that your manuscript has been deemed suitable for publication in PLOS ONE. Congratulations! Your manuscript is now being handed over to our production team.

Kind regards, 

on behalf of

Prof Novel Njweipi Chegou 

Academic Editor

PLOS ONE